# Medical students impacted by discrimination: a qualitative study into their experiences of belonging and support systems at medical schools in the UK

Hamza Ikhlaq [ORCID],[1] Srishti Agarwal [ORCID],[2] Catharine Kwok [ORCID],[2] Hadiya Golamgouse,[2] Simone Derby,[3] Nicola McRae [ORCID],[4] Megan E L Brown [ORCID],[5] Victoria Collin [ORCID],[5] Ravi Parekh [ORCID],[5] Sonia Kumar[6]

¹Faculty of Medicine, Imperial College London, London, UK
²University College London, London, UK
³University of Nottingham, Nottingham, UK
⁴University of Dundee, Dundee, UK
⁵Medical Education Innovation & Research Centre, School of Public Health, Imperial College London, London, UK
⁶University of Leeds, Leeds, UK

**Correspondence to**
Mr Hamza Ikhlaq;
hamza.ikhlaq18@imperial.ac.uk

## ABSTRACT

**Objective** To better understand the broader experience of medical students impacted by discrimination and the support systems they engage with.

**Design** Qualitative study using semi-structured interviews.

**Setting** Four medical schools based in the UK.

**Participants** 17 medical students were recruited using volunteer and snowball sampling: all students self-identified as being impacted by discrimination.

**Results** 5 themes were identified: feelings of isolation, imposter syndrome and exclusion; a lack of representation and positive role modelling; the importance of peer support; issues relating to the accessibility of support; building support networks through shared experiences and attempts to foster a sense of inclusion through peer and institutionally led initiatives.

**Conclusions** The findings of this study suggest medical schools could do more to recognise the importance of acknowledging the multiple identities at risk of discrimination held by students, perpetuating feelings of isolation and exclusion. Our research highlights the need for practical systemic initiatives to improve the sense of belonging of medical students who are impacted by discrimination. Medical educators and institutions should consider formal and informal provisions, such as creating time and space for students to meet and share experiences, access support and reporting networks, to foster a greater sense of belonging.

## STRENGTHS AND LIMITATIONS OF THIS STUDY

⇒ This study used a novel method of exploring peer perspectives via a team largely composed of medical student researchers, who themselves have lived experiences of discrimination within medical school.

⇒ Our research is unique in its approach as it examines student belonging from a broader perspective, hearing from those impacted by discrimination and acknowledging the multiple identities that students can have.

⇒ Although this study has a small sample, it captures the range of experiences across four institutions and across multiple years of medical school programmes.

⇒ Given the complexity and sensitivity of discrimination in medical school, semi-structured interviews have allowed us to explore the personal and often sensitive experiences of participants.

⇒ This multi-institutional study covers 4 of the 47 current medical schools in the UK. Therefore, there is scope for further examining the experiences of students with intersecting identities and experiences across varying geographical locations within the UK and beyond.

## INTRODUCTION

To experience discrimination is to be treated unfairly or with prejudice based on the characteristics of a group, or an individual.[1] Strayhorn[2] defined a sense of belonging within higher education as:

‘students' perceived social support on campus, a feeling or sensation of connectedness, and the experience of mattering or feeling cared about, accepted, respected, valued by, and important to the campus community or others on campus such as faculty, staff, and peers.' (p.4).

Belonging has been linked to positive academic outcomes, high engagement and self-confidence among university students[3] and is pertinent in students who are impacted by discrimination as they are already susceptible to exclusion and isolation.[4] Such students who have reported mistreatment and discrimination in medical school, particularly those in their earlier years, have subsequently been



found to be more likely to discontinue from medical school.[5]

Historically, the makeup of medical students in the UK is skewed towards students from more affluent backgrounds.[6] As a result, UK medical schools are now obliged to have schemes that ensure medicine is accessible to a wider population.[7] Widening participation (WP) is an example of such an approach and is defined as: 'ensuring that students from disadvantaged backgrounds can access higher education, get the support they need to succeed in their studies, and progress to further study and/or employment suited to their qualifications and potential'.[6] However, students impacted by discrimination, whether it be due to their WP status or those at risk of societal discrimination due to minoritised characteristics, have been found to face barriers in every stage of their medical careers.[8]

According to Jackson, 'Fairness and equality in medical education is not just a problem at the front door; it continues throughout medical school.'[9] Current literature primarily focuses on interventions for post-16 students to boost enrolment into medical school. However, there is limited research into the follow-up of students impacted by discrimination during their time in medical school.[10] There is emerging evidence that students facing discrimination have different learning and social experiences and further research is required to address systemic issues within medical schools.[11] Indeed, the prescriptive nature of simply increasing the number of students from disadvantaged backgrounds into medical schools has a risk of 'tokenising' these students and further alienating their sense of belonging.[12] Medical students subject to discrimination can experience social exclusion from the wider cohort and inequality in accessing resources. This can result in a diminished sense of trust between these students and the institution.[13] There are also specific barriers such as fears of stigmatisation and a lack of accessibility for neurodivergent students,[14 15] which influence a poorer sense of belonging due to poor relationships with the institution, staff and clinicians.[13] Though important, and interesting, these studies are limited as they typically focus on exploring one group of student's sense of belonging, such as ethnically minoritised students or socioeconomically disadvantaged populations.

There is a need for further research to examine the wider experiences of medical students impacted by discrimination and the intersections of identities for students who may have two or more characteristics that are at risk of discrimination. Furthermore, it is imperative to understand what support these students would require from their universities to maximise student engagement. Therefore, this project examined the experiences of medical students who have been impacted by discrimination, and the support systems available to them. We aim to generate practical recommendations to improve systemic and holistic provisions for these students, thereby enhancing their sense of belonging.

## METHODS
Given the complexity and sensitivity of discrimination in medical school, we chose to conduct semi-structured interviews to explore the thoughts, beliefs and experiences of participants. We created 12 questions to form the basis of our semi-structured interviews covering various themes of belonging, networks, support systems and experiences of discrimination using findings from our literature review. Each participant was interviewed individually (online supplemental appendix 1). We used a constructivist paradigm[16] using an interpretative description approach.[17] This enabled us to fully explore each participant's subjective experiences and understand the influences behind these, in order to develop practical recommendations to improve belonging and support systems at medical school. Data were collected between October and December 2022. All participants were emailed welfare debriefing information specific to their institution and SK was the assigned point of contact in the event of serious concern.

### Sample and recruitment
Medical students who self-identified as having been impacted by discrimination were eligible to take part in the study. Students who had just undertaken an intercalated BSc were excluded from the study, as they may be affiliated with a department external to the medical school and not all medical students may have the opportunity to intercalate. Participants self-identified as having been impacted by discrimination using an existing, broadened WP definition.[18] This allowed us to collect a diverse multitude of student experiences, which may be currently underrepresented in existing literature and policies. Our recruitment text provided some guidance through examples of discriminated groups including but not limited to:

► Those who identify as being otherly abled/having a disability/receiving Disabled Students' Allowance (DSA).
► Those who identify as an ethnic minority group.
► Those who identify as lesbian, gay, bisexual, transgender, queer, questioning, intersex, or asexual (LGBTQIA+).
► First-generation applicants to higher education.
► Those previously looked after by a local authority.
► Those who are themselves or have parents/guardians in receipt of means-tested benefits.
► Those with household addresses in a participation of local areas (POLAR) classification 3 or 4 area.
► Those with refugee status.
► Young carers.
► Students who belong to one of the priority engagement groups (includes students from Gypsy, Roma and Traveller communities and children in military families).
► Those who are estranged from both parents/legal guardians during their higher education.
► Mature students.

For the full recruitment text, see online supplemental appendix 2.

We recruited through the host institution using email, newsletters and relevant social media platforms. Adverts were sent out through these platforms asking students to sign up. These students were then contacted by the research team to check eligibility and availability for an interview.

In addition to fulfilling our recruitment criteria, participants were eligible to participate if they were available for an interview and provided consent to participate. We aimed to ensure breadth of experiences, maximising variability across year groups; for each university, we selected one student per year group. All participants were compensated for their time through £20 food vouchers. In total, we recruited 17 participants from all years of the medical school programmes across four UK-based medical schools using convenience sampling. This allowed us to achieve our study aims and ensure a breadth of experiences, exploring their experiences to conclude their sense of belonging and experience of support at medical school.

## Procedure

After selection, each participant was invited to a 60-minute semi-structured interview. Each interview was run by two co-investigators (HI, SA, CK, HG, NM and SD), with one leading and one supporting the interview (prompting additional questions to explore students' experiences or in case of technical difficulties). Consent was gained prior to the interview through the submission of an online consent form and verbal consent to record the session was obtained prior to the start of each discussion. To avoid potential conflicts of interest, investigators only interviewed participants external to their own institutions.

## Data analysis

After transcription, we anonymised data and followed a constructivist approach to qualitative thematic analysis. We analysed data inductively and followed Braun and Clarke's[16] six steps: familiarisation; data coding; generating initial themes; reviewing and developing themes; refining, defining and naming themes; and writing the report. We developed a code list, which all members of our team reviewed. At least one member of our team peer-reviewed each transcript to add depth to coding (HI, SA, CK, HG, NM and SD). Throughout the analysis, we modified existing codes and created new codes through discussion. We then connected these codes to develop themes through team discussions. All data analyses were conducted using the Dedoose (V.9.0) software.

## Patient and public involvement

Patients or the public were not involved in the design or conduct of this study.

## Reflexivity

The research team comes from varying ethnically minoritised backgrounds and is predominantly female identifying, all with a strong interest in medical education. All student researchers identify as having been impacted by discrimination, thus leading the research team to explore this research project and have a strong aspiration to make positive recommendations for medical schools. Due to the nature of our study and personal experiences of discrimination, peer debriefing through online meetings were used to reflect and deepen our understanding of recounted experiences shared by study participants and provide support if sensitive topics arose.

## RESULTS

Five themes were identified from the 17 interviews conducted: feelings of exclusion; failures of organised support systems; representation within the medical school curriculum, staff and peers; the importance of peer support and attempts to foster a sense of inclusion. A summary of the characteristics of our study population is provided in table 1.

### Feelings of exclusion

One of the main barriers to achieving a sense of belonging experienced by participants was feelings of exclusion at an individual and institutional level. At an individual level, minoritised characteristics such as ethnicity, socioeconomic class, religious beliefs and caring responsibilities were key factors which participants felt perpetuated feelings of exclusion. For example, one participant highlighted the lack of diversity among peers as the reason they 'never feel [like they] fully belong' (P012). This was further exacerbated by intersections of identity as 'being an ethnic minority wearing a hijab' made them feel they were 'a minority within a minority' (P012). Opportunities to socialise with peers sometimes conflicted with participants' cultural and religious beliefs where 'every single event centred around drinking' with a lack of alternative events available (P005). Furthermore, participants highlighted when feelings of exclusion were felt it was because they 'couldn't connect' to their peers due to their unique experiences, such as 'experiences as a care leaver', thus resulting in self 'blame' to explain the 'disconnect' (P002).

At an institutional level, participants expressed a disconnect between the medical schools and students. One student felt that they felt they were 'set up to fail' in a system which does not recognise their intersection of identities, thus 'disempowering' them (P008). Feelings of exclusion echoed into clinical attachments where students felt senior doctors would make 'split-second' judgements based on 'prejudice', to the extent they felt 'belittled' (P012). Examples of this emerged particularly in surgical theatres where students who wore hijab felt 'discouraged from going to theatre' (P012).

### Failures of support systems

Participants had experienced failings in the formal support networks and structures provided by their medical schools. Fixed structures, such as requiring a

**Table 1** Characteristics of our study population

| Participant | Gender | Medical school | Year group in the academic year 2021–2022 |
|---|---|---|---|
| P001 | Male | Imperial College London | 1 |
| P002 | Non-binary | Imperial College London | 2 |
| P003 | Male | Imperial College London | 3 |
| P004 | Non-binary | Imperial College London | 4 |
| P005 | Female | Imperial College London | 5 |
| P006 | Female | University College London | 1 |
| P007 | Female | University College London | 2 |
| P008 | Male | University College London | 3 |
| P008 | Female | University College London | 4 |
| P009 | Male | University of Nottingham and Lincoln Medical Schools | 1 |
| P010 | Female | University of Nottingham and Lincoln Medical Schools | 2 |
| P011 | Female | University of Nottingham and Lincoln Medical Schools | 3 |
| P012 | Female | University of Nottingham and Lincoln Medical Schools | 4 |
| P013 | Female | University of Dundee | 1 |
| P015 | Female | University of Dundee | 2 |
| P016 | Female | University of Dundee | 3 |
| P017 | Female | University of Dundee | 4 |

tutor referral for welfare support made it 'very difficult… to access other welfare support services' (P002) when their tutor had been uncontactable for an extended period of time. This barrier to accessing pastoral support was further demonstrated by another participant questioning: 'how desperate are you to go through all … those hoops to actually get the support?' (P004).

Other failings have included experiences of being 'bounced between people and no one really is quite

sure what's happening' (P008). Consequently, this 'one-size-fits-all' (P003) approach has built a feeling of lack of genuine care: 'they want to tick a box saying they've provided support' (P012) and one participant expressed that they had 'a level of distrust … within the system' (P008). Lack of follow-up and ongoing support was evidenced: 'it almost feels prescriptive, that's just like okay, we've met, great, we've done our job, you must be better now' (P004).

## Representation
Representation in medical school was a common topic that was raised as a source of comfort, helping to improve belonging, and equally, the lack thereof challenged participants.

Having an open space to discuss identities, alongside others similarly identifying individuals helped provide a sense of familiarity to participants, namely for LGBTQ+ individuals who struggled to find this in the 'heteronormative blanket' (P008). Additionally, the diversity of faculty members also greatly helps to improve belonging, with one participant 'so genuinely happy to see a hijabi lecturer because they were the only one I've seen' (P012).

Issues with representation arose on several occasions, such as the integral makeup of students with 'most' of the student population being educated in 'private schools' and the class divide that inherently exists between these students and students impacted by discrimination (P012). Furthermore, participants felt they may be represented in faculty as a 'woman' but did not feel the same with regard to their 'ethnicity'. (P013) Additionally, the lack of inclusive teaching, including signs and skin pathology in ethnic minority patients, was raised twice (P012, P013) as a significant problem that impacts belonging. Indeed, the '(lack of teaching) makes you feel like *you* don't matter, *you're not important.*' (P012)

Despite this, one student did raise some improvements that have been made, which have improved their sense of belonging through representation. There have been more 'active efforts to embrace people of different backgrounds' in clinical and academic environments which have been 'successful' and have increased their sense of belonging (P003).

## Peer support
Some participants described how peer support, resulting from positive interpersonal relationships between themselves and members of their cohort, and the wider university population has impacted their experiences at medical school as students who have experienced discrimination. For those who have very little or no peer support, there was a desire to have that access. Having friends to help provide support was said to be integral in providing 'emotional padding' (P002) which some students have found was often omitted from their lives up to this point. To some students, the willingness of others to talk 'openly and give support' is a 'big deal' (P003), especially for

those who are going through what seems to be challenging times.

However, participants specified preferences that would help maximise the efficiency of peers as a support system which included peers who are 'of older years' (P002) and/ or with whom they 'have that shared experience' (P001). This stems from the notion that as much as 'peers who aren't from that background try to empathise and try to relate…they don't have the lived experience' (P003), which creates a barrier to feeling understood and supported.

### Feelings of inclusion

Participants tended to express feelings of inclusion when reflecting on their experiences with communities that they 'had a lot in common with' (P002). The freedom to be who you are without judgement was important to most students who found comfort in diverse student groups ranging from sports to LGBTQ+ societies. Feeling like 'something bigger than yourself' gave students the strength to 'be sort of whatever you want to be' (P003) which in turn improved their overall sense of belonging.

However, feeling included was a process that required active effort, as one participant expressed, 'I've had to fight to think that I belong' (P003). Several participants changed elements of themselves to better fit in, which only added to feeling more out of place. Feelings of inclusion were also constantly challenged by themselves, by their peers, and by the medical school throughout their time at university. Imposter syndrome was attributed to this challenge. Students struggled to keep up with their peers and thought they weren't 'smart enough' (P008) to be at medical school. The lack of support at this level meant that students held onto these beliefs until they found a community which gave them 'the confidence to do well, but also equally fail' (P003).

Additionally, several students considered loneliness when reflecting on their experiences of inclusion. Many described feeling isolated from their peers due to characteristics largely outside of their control such as gender and sexuality. Despite overcoming several hurdles to reach medical school, participants described feeling as if they alone did not belong and that they were out of place. As stated by P002 'for me, I took most of my first year of medical school thinking "oh my god, I'm the only one" that everyone else was 'normal'. The importance of community was therefore highlighted in all participant reflections and was explored as a primary cause for improved feelings of inclusion.

### DISCUSSION

Through semi-structured interviews, we explored the perceived sense of belonging felt by medical students across four UK medical schools who have been impacted by discrimination and the support systems they have accessed. Our study highlights how feelings of exclusion impacts medical students' experiences whilst at medical school. Students from minoritised backgrounds reported a lack of peer diversity and understanding of their experiences from peers. Furthermore, barriers to engaging in accessible social and professional activities often impacted participants' self-esteem.

Diversity in student makeup is associated with perceived enhancements in educational experiences.[19] Furthermore, works by Machado *et al* have suggested socioeconomic and racial diversity offers opportunities to reform medical culture, but student socialisation must be supported through activities to support self-esteem.[20] Students also described experiences of marginalisation and stereotyping in clinical attachments, exacerbated by the professional hierarchy, which has been linked to moral distress, trainee mistreatment and amplification of fatigue, stress and burnout.[21] As well as this being reflected in our study, we find feelings of self-blame and a resultant lack of self-worth are sometimes used by students to explain the disconnect between their experiences and those of their non-marginalised peers, further disempowering them from fully engaging with clinical, non-clinical and social experiences provided within medical school.

The need for representation extends within the formal curricula as participants highlighted the lack of inclusive teaching on disease in ethnically minoritised patients. Considering the topics discussed above, it is perhaps not surprising that participants felt displaced when teaching solely focused on white skin, with signs and symptoms only noted in those belonging to said populations.[22] Participants in this study highlighted how seeing themselves represented within the curriculum allowed them to feel a greater sense of self-importance and observing active institutional efforts to diversify the curriculum fostered positive feelings towards their respective medical schools. Thus, representation, both in the physical form of a diverse and intersectional cohort of staff and doctors, coupled with inclusive, non-stereotyped, teaching and the provision of allocated spaces to discuss these pertinent issues, would greatly improve the sense of belonging for students impacted by discrimination.

Students reported experiences of barriers to accessing support through often hard-to-reach, unavailable networks or outdated systems. The ease of access is a key area for institutions to focus on, for example, by providing multiple access points.[23] Institutions should be proactively encouraging students to seek support at an earlier and more manageable stage, ensuring that follow-up opportunities are integrated. We have demonstrated that when a 'one-size-fits-all' and tokenistic system of support disappoints an individual, students feel isolated and cynical. Providing individualised support tailored to appropriately match the diverse identities of the student population is paramount in order to bridge this disconnect between stated and perceived institutional values.[24] We have demonstrated this is vital in ensuring students can build authentic connections with support staff to establish trust and reduce the confusion felt by students when attempting to access relevant support services. However,

to implement such support, pastoral support needs to be prioritised and appropriately renumerated to overcome logistical barriers, such as time and financial costs.

This research has shown that across cohorts, medical students who have experienced discrimination desire peer-to-peer support to help improve their sense of belonging. However, this peer support must be done in a way in which students feel will be most beneficial, such as from older students, those considered friends and those with a shared experience. Peer-to-peer support has been shown to help medical students who may feel burnt out, inadequate, socially isolated and anxious,[25] and is found to have positive impacts on relationships with peers as they become professionals.[26] The development of peer support systems that are established in partnership with those students who have had shared experiences, while useful, should be mindful not to place all the responsibilities for this on an overburdened cohort of students.

Many medical students avoid seeking help, even more so when the university is involved, due to the implications they believe this will have on their futures as working professionals.[27] This phenomenon in combination with the results above, highlights that while medical schools have specialist support systems in place to support students, the support that is desired from students is from their peers. Our study highlights the need for medical schools to implement programmes within the degree that help equip their students with the tools to best support their peers within their capacity. We have also highlighted the need for medical schools to help ensure there is a bridge between students across different year groups. This could be done through schemes such as mentoring pairings between older and younger years, ensuring a degree of compatibility between the mentor and mentee so that they have those shared or similar experiences.

Table 2 provides a summary of our findings and explores potential recommendations for medical schools to help improve the sense of belonging for students impacted by discrimination.

## STRENGTHS AND LIMITATIONS

Previous research typically focuses on exploring belonging from a generic student perspective or from students belonging to a specific socio-demographic background. Instead, our research examined students belonging from a broader perspective, hearing from those impacted by discrimination and acknowledging the multiple identities that students can have. This study used a novel method of exploring peer perspectives via a team largely composed of medical student researchers, whom themselves have lived experiences of discrimination within medical school. The student-led nature of this research allowed for a deeper and clearer understanding of student experiences. Furthermore, our research was able to meet the criteria for conceptual depth by having a range of evidence and removing any need to make assumptions through the use of our in-depth semi-structured interviews.[28] Our research

**Table 2** A summary of the key findings of this study and a summary of recommendations for medical schools to help improve the sense of belonging experiences by students impacted by discrimination

| Findings | Recommendations for medical schools |
|---|---|
| Feelings of exclusion, imposter syndrome and low self-worth are common among students experiencing discrimination. | Workshops at the start of medical school to raise awareness of imposter syndrome and its prevalence within the medical school and the possible implementation of support for students, potentially through peer mentors and staff, for students who possess characteristics at risk of discrimination. We can perhaps further empower those who have these experiences to work in partnership with institutions in designing these workshops and where appropriate are rewarded for their time in remuneration or recognition. |
| Participants desire a sense of community to meet like-minded individuals. | Networking and peer social events for students to meet and discuss ideas and share experiences. |
| Feelings of inclusion can be fostered through representative teaching and the diversity observed among staff. | Ensure teaching considers the social and cultural diversity of the student body and ensure diversity among the student-facing workforce. |

not only contributes to the knowledge in this area by providing information on the personal experiences of medical students impacted by discrimination but importantly highlights areas for change and recommendations for support that could be offered by universities that may improve the sense of belonging for these students.

Nevertheless, this multi-institutional study covers only 4 of the 47 current medical schools in the UK. Therefore, further research is required to fully explore each medical school's support structure for medical students impacted by discrimination, such as further examining the experiences of students with intersecting identities and the experiences across varying geographical locations within the UK and beyond. There is the possibility of self-selection bias among the 17 participants; however, the self-identification inclusion criteria were broad and students were recruited across multiple year groups and medical schools. Although the cross-sectional design of this study yielded novel findings, further longitudinal studies are warranted to examine the long-term consequences for students impacted by discrimination, particularly as they progress throughout medical school and into their careers. Indeed, although beyond the scope of this paper, it would be interesting for future works to examine

the differences in the sense of belonging felt outside of medical school compared with those experiences while in medical school. Furthermore, the heterogeneous nature of medical programmes across institutions is something that must be accounted for when considering the experiences of belonging and implementations for change. We would like to further explore these recommendations by taking the findings of this research to the faculty leaders in these institutions and other medical school bodies to stimulate further discussion on what strategies they may implement to improve these students' sense of belonging.

## CONCLUSION

In this study, we explored the experiences of medical students impacted by discrimination, their sense of belonging and the support systems they accessed. The findings of this study suggest positive interpersonal relationships among peers and those with similar lived experiences can provide emotional safety and support for students to feel more included. However, there remains a significant lack of social and cultural awareness among the student body and support structures that may perpetuate feelings of isolation and exclusion. Furthermore, the lack of representation can drive feelings of a reduced sense of belonging and values of self-belief among students. To improve, medical educators and institutions must consider inclusive, representative curricula and diversity among their student-facing staff. Moreover, there is a need for medical schools to improve their support and recognition of the multiple identities held by students through practical systemic initiatives. This includes both formal and informal provisions offered to students to build communities and networks among peers, alongside accessible, personalised support pathways, thereby allowing students impacted by discrimination to foster a greater sense of belonging.

**Acknowledgements** We would like to thank all the students who volunteered as participants in this study. We hope your stories will continue to better the future of medical education.

**Contributors** HI is the first author and the guarantor of this study. HI, SA, CK, HG, SD and NM contributed to the study conception, protocol development, data collection, data interpretation and critical revision of the manuscript. MB, VC, RP and SK provided supervising guidance and support utilising their expertise in the field of medical education.

**Funding** This article presents independent research commissioned by the National Institute for Health Research (NIHR) under the Applied Health Research (ARC) programme for North West London. The open access fee was paid from the Imperial College London Open Access Fund.

**Disclaimer** The views expressed in this publication are those of the author(s) and not necessarily those of the NHS, the NIHR or the Department of Health and Social Care.

**Competing interests** None declared.

**Patient and public involvement** Patients and/or the public were not involved in the design, or conduct, or reporting, or dissemination plans of this research.

**Patient consent for publication** Not applicable.

**Ethics approval** This study involves human participants and ethical approval was given by the Educational Ethics Research Committee at Imperial College London, with reciprocal approval granted by partner medical schools of the research team (reference number: EERP2122-068a). Participants gave informed consent to participate in the study before taking part.

**Provenance and peer review** Not commissioned; externally peer reviewed.

**Data availability statement** All data relevant to the study are included in the article or uploaded as online supplemental information.

**ORCID iDs**
Hamza Ikhlaq http://orcid.org/0000-0002-5834-5728
Srishti Agarwal http://orcid.org/0009-0005-9569-9524
Catharine Kwok http://orcid.org/0009-0002-0293-3659
Nicola McRae http://orcid.org/0009-0008-0202-2680
Megan E L Brown http://orcid.org/0000-0002-9334-0922
Victoria Collin http://orcid.org/0000-0003-0032-1712
Ravi Parekh http://orcid.org/0000-0003-0219-4956

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
