## [Reviewer comments · BMJ Open]

ARTICLE DETAILS

TITLE (PROVISIONAL)	Medical students impacted by discrimination: a qualitative study into their experiences of belonging and support systems at medical schools in the United Kingdom
AUTHORS	Ikhlaq, Hamza; Agarwal, Srishti; Kwok, Catharine; Golamgouse, Hadiya; Derby, Simone; McRae, Nicola; Brown, Megan; Collin, Victoria; Parekh, Ravi; Kumar, Sonia

VERSION 1 – REVIEW

REVIEWER	Diaz, Adrian Ohio State University, General Surgery
REVIEW RETURNED	16-Sep-2023

GENERAL COMMENTS	In this qualitative study the authors seek to better understand the broader experience of medical students impacted by discrimination. They utilize semi-structured interviews and analyze the transcripts using a constructivist approach to a thematic analysis. The authors conclude that medical schools are failing to recognize the importance of acknowledging the multiple identities at risk of discrimination held by students, perpetuating feelings of isolation and exclusion. This is a timely study that is well executed. I have several comments for the authors, that I hope will help strengthen their study. 1. The authors conclude that “medical schools are failing to recognize the importance of acknowledging the multiple identities at risk of discrimination held by students, perpetuating feelings of isolation and exclusion”. Can the authors highlight some examples where schools are not failing students in these domains? Were there any examples provided from the interviews, and if so they should be highlighted in the study. If there were no examples, I think it would be helpful for the authors to perhaps pontificate how medical schools can improve. Rather than framing this study as “medical schools failing”, I think it would be more useful to the reader and to medical school administrators to ready a study framed as “this is how medical schools can improve”2. the introduction is a bit lengthy and can probably be significantly shortened without losing its meaning For example, the last paragraph can be eliminated altogether as its described in the methods section.3. The methods are well described, and appropriate references are provided. My only concern is that of generalizability. Only 17 medical students are interviewed from 4 medical schools. How confident are the authors that thematic saturation was reached?
---

	4. Additionally, how did the authors address self-selection bias. I must imagine that students who decided to participate must have been feeling some sort of discrimination or neglect from medical schools. How can we be sure that these findings are shared amongst most students and not only the 17 who self-identified to be interviewed. In other words, do the authors believe that the same themes would arise if another 17 students were chosen at random? 5. the straighten and limitations are addressed well. The conclusion can proabbaly be condensed to one paragraph.
--	--

REVIEWER	Najjar, Iris HUG, Medicine
REVIEW RETURNED	29-Oct-2023

GENERAL COMMENTS	Hello, Thank you for your initiative, I think we have to give more importance to this intersectional approach. Because of the limited number of participants, I would have appreciated a more in-depth analysis of the discrimination experience. Maybe give more concrete examples, talk about the long-term consequences (5th year med students..). Also, the question that arises for me is : how is their experience of discrimination different in medical school than outside of school? Indeed often, these same people also experience discrimination outside school and it would be interesting to know if there is something particular to med school that enhances this experience. It would also be interesting to compare with other types of studies... I would change this sentence as a feeling of exclusion implies a lack of belonging : Our study highlights how feelings of exclusion prevent medical students from achieving a sense of belonging.
---

VERSION 1 – AUTHOR RESPONSE

Reviewer 1

Comment 1: The reviewer expressed concern regarding the framing of medical schools as “failing” as they perceived a lack of specific examples from the interviews. They advised that we reframe the conclusion to highlight improvements that could be made, which would be more useful to readers and medical school administrators.

Response: This suggestion has been accepted. We have reframed our conclusion to suggest medical schools could improve their systematic support of students impacted by discrimination, and ways in which they could do this.

Comment 2: The reviewer highlighted the introduction as being lengthy and suggested shortening it, which could be done without losing meaning.

Response: Thank you for this suggestion. We have now deleted the last paragraph as per the reviewer's suggestion. The ending of the introduction has now been condensed.

Comment 3: The reviewer expressed concern regarding generalisability of our study and our confidence that theme saturation was reached.

Response: Thank you for making us aware of this concern. We looked at geographically dispersed medical schools, had a diverse range of criteria reached by participants, and different cohorts with different experiences. This allowed for a range of evidence which was coded and resonated with what current literature highlights. The use of in-depth semi-structured interviews allowed us to interrogate what participants said in real-time, removing the need to make any assumptions based on any phrasing that may be ambiguous or multi-layered. This in turn allowed for our study to meet the requirements for conceptual depth, which we found to be most appropriate for our study which was qualitative in nature, in comparison to theme saturation (1). We were able to see and appreciate what is done well to support medical students, and also form recommendations for medical schools based on recurring themes of what could be done better, what has been done well, or not done at all. This has now been added into the strengths and limitations section.

Comment 4: The reviewer wished to know more about self-selection bias and if we believe the same themes would arise if another 17 students were chosen at random.

Response: Thank you for identifying this potential bias; we have now addressed this in our limitations section. Our study aim was to explore the experiences of students who had been impacted by discrimination; as such, the aspect of self-selection is not necessarily creating bias in our findings, but providing us the insights we were seeking. Though we had 17 students' experiences, these were varied by year group and institution and we provided broad inclusion criteria with self-identification. We believe that this minimised the biases, though future research with a larger sample size could further prove this.

Comment 5: It was suggested that the conclusion could be condensed to one paragraph.

Response: The conclusion has now been revised and condensed to one paragraph.

Reviewer 2

Comment 1: The reviewer wished to see more in-depth analysis of the discrimination experience, including long-term consequences.

Response: We thank the reviewer for this point. Due to the aims of the study to capture the range of experiences of students impacted by discrimination, quotes have been embedded throughout the results to exemplify the themes generated through the constructivist approach utilised. However, we acknowledge the limitations of our cross-sectional study and we have added in the need for future longitudinal studies to examine long-term consequences for students in the limitations section.

Comment 2: The reviewer expressed an interest in reading how their experience of discrimination outside of medical school differ to their experience within medical school. They also suggested comparison to other studies to identify if there is something particular to medical school that exacerbates the discrimination experience.

Response: We thank the reviewer for this insightful point. Although interesting, we feel this is beyond the scope of our study. The study examines the experiences of belonging in situ, whilst students are within medical school and aimed to generate recommendations for higher education institutions. Nevertheless, we acknowledge this is an interesting point and further studies could examine such experiences outside of medical schools to identify exacerbating factors. This has now been added into the limitations and exploration for future work.

Comment 3: It was suggested to alter the following sentence, as it implied a lack of belonging: "*Our study highlights how feelings of exclusion prevent medical students from achieving a sense of belonging.*"

Response: This suggestion has been accepted, and the sentence has been altered to the following: “Our study highlights how feelings of exclusion impact medical students’ experience at medical school.”

References

1. Nelson, J. Using conceptual depth criteria: addressing the challenge of reaching saturation in qualitative research. *Qualitative Research* 2017; 17(5):554-570. Doi: 10.1177/1468794116679873

VERSION 2 – REVIEW

REVIEWER	Diaz, Adrian Ohio State University, General Surgery
REVIEW RETURNED	28-Nov-2023
GENERAL COMMENTS	The manuscript has been adequately revised, addressing all of my comments (and other reviewer comments).